# Stage Shifting by Modifying the Determinants of Breast Cancer Stage at Diagnosis: A Simulation Study

**DOI:** 10.3390/cancers16061201

**Published:** 2024-03-19

**Authors:** Gyanendra Pokharel, Qinggang Wang, Momtafin Khan, Paula J. Robson, Lorraine Shack, Karen A. Kopciuk

**Affiliations:** 1Department of Mathematics and Statistics, University of Winnipeg, 515 Portage Avenue, Winnipeg, MB R3B 2E9, Canada; g.pokharel@uwinnipeg.ca; 2Cancer Epidemiology and Prevention Research, Cancer Care Alberta, Alberta Health Services, 2210 2nd Street SW, Calgary, AB T2S 3C3, Canada; qinggang.wang@albertahealthservices.ca (Q.W.);; 3Department of Agricultural, Food and Nutritional Science and School of Public Health, University of Alberta, 116 Street & 85 Avenue, Edmonton, AB T6G 2P5, Canada; paula.robson@albertahealthservices.ca; 4Cancer Care Alberta and Cancer Strategic Clinical Network, Alberta Health Services, 10030-107 Street NW, Edmonton, AB T5J 3H1, Canada; 5Cancer Surveillance and Reporting, Cancer Care Alberta, Alberta Health Services, 2210 2nd Street SW, Calgary, AB T2S 3C3, Canada; lorraine.shack@albertahealthservices.ca; 6Department of Oncology, University of Calgary, 1331 29 Street NW, Calgary, AB T2N 4N2, Canada; 7Departments of Community Health Sciences and Mathematics and Statistics, University of Calgary, 2500 University Drive NW, Calgary, AB T2N 1N4, Canada

**Keywords:** Alberta’s Tomorrow Project, breast cancer stage, breast neoplasms, early detection of cancer, lifestyle, neoplasm staging, simulation study, stage shift

## Abstract

**Simple Summary:**

The impact of breast cancer on individuals and in populations can be reduced if the cancer is detected in an early stage of disease as treatments are generally less harsh and more successful. Knowing which factors to modify and by what amount that could shift breast cancer diagnoses to early stages of disease were studied in this project. Factors related to stage at diagnosis in a previous analysis of women diagnosed with breast cancer were assessed using data that mimicked the real data. The proportions of modifiable lifestyle factors in a population were increased or decreased, such as those ever having a mammogram, to see the effect on detecting more cancers in the earliest stages. Similarly, the average amounts of protein eaten per day or calories consumed were also increased and decreased. Increasing the total amount of protein eaten per day by even just 5 grams was the most important factor found in this study to improve early breast cancer diagnoses. The findings from this study could be helpful in breast cancer prevention programs that could target key modifiable factors and by other researchers studying other types of cancer.

**Abstract:**

Background: Breast cancer is the most common cancer in Canadian women; nearly 25% of women diagnosed with cancer have breast cancer. The early detection of breast cancer is a major challenge because tumours often grow without causing symptom. The diagnosis of breast cancer at an early stage (stages I and II) improves survival outcomes because treatments are more effective and better tolerated. To better inform the prevention of and screening for breast cancer, simulations using modifiable rather than non-modifiable risk factors may be helpful in shifting the stage at diagnosis downward. Methods: Breast cancer stages were simulated using the data distributions from Alberta’s Tomorrow Project participants who developed breast cancer. Using multivariable partial proportional odds regression models, modifiable lifestyle factors associated with the stage of cancer at diagnosis were evaluated. The proportions or mean levels of these lifestyle factors in the simulated population were systematically changed, then multiplied by their corresponding estimated odds ratios from the real data example. The effects of these changes were evaluated singly as well as cumulatively. Results: Increasing total dietary protein (g/day) intake was the single most important lifestyle factor in shifting the breast cancer stage downwards followed by decreasing total dietary energy intake (kcal/day). Increasing the proportion of women who spend time in the sun between 11 am and 4 pm in the summer months, who have had a mammogram, who have been pregnant or reducing the proportion who are in stressful situations had much smaller effects. The percentage of Stage I diagnoses could be increased by approximately 12% with small modifications of these lifestyle factors. Conclusion: Shifting the breast cancer stage at diagnosis of a population may be achieved through changes to lifestyle factors. This proof of principle study that evaluated multiple factors associated with the stage at diagnosis in a population can be expanded to other cancers as well, providing opportunities for cancer prevention programs to target specific factors and identify populations at higher risk.

## 1. Introduction

Breast cancer mortality has been declining at a steady rate in women since 1988 [1], primarily due to early detection and enhanced treatment options [1,2]. Still, 13% of all cancer deaths in Canada are breast-cancer-related, many of which might have been averted had they been diagnosed at an earlier stage [1]. Breast cancer stages are based on tumour characteristics, including size (T), lymph node involvement (N) and spread (metastases, M) to other tissues [3]. Breast cancer stages range from 0 to IV, with early-stage disease defined as stages I and II [4]. Early-stage disease at diagnosis confers substantial survival benefits because treatments are more effective and better tolerated [2]. The five-year net survival rates for Canadian females diagnosed at stage I is 99.8% but decreases to 74.0% for those diagnosed at stage III and only 23.2% at stage IV [4]. Since the primary prevention of breast cancer is not always possible, the detection of early-stage disease is critically important for cancer control [2,5].

Previous studies have examined delays in cancer diagnoses at a population or neighbourhood level, including effects of ethnicity and socioeconomic status [6]. A Canadian report on population-level factors showed that the likelihood of early cancer diagnosis differed among income groups, geographical areas, and immigration status [7]. Other Canadian studies also found lower annual mammography screening participation for women with low socioeconomic status despite free access to breast cancer screening programs for age-eligible women [8,9]. Organized breast cancer screening programs have led to more women being diagnosed at stages I and II (82%); however, there is still a significant proportion of breast cancer patients diagnosed in later stages [10]. Other reasons for late-stage diagnosis may involve individual-level factors, including health behaviours, lifestyle factors and health insurance coverage [6,11,12,13]. A prospective healthy cohort, such as Alberta’s Tomorrow Project (ATP), is the ideal study design to investigate many individual-level factors associated with stage at diagnosis [14]. Access to free breast cancer screening for women aged 50–74 (the screening program eligibility age range at time of diagnosis) province-wide in Alberta, Canada reduces the barriers to healthcare access that have been identified in other studies. 

The majority of research on individual-level factors associated with delayed diagnosis of breast cancer has found that a history of breast cancer screening decreases the risk of late-stage diagnosis, but other factors, such as body mass index (BMI), smoking, physical activity, diet, distance from health centre, pregnancy and comorbidity, have inconsistent associations [13,15,16]. Discrepancies between previous research studies may be explained by different methods of statistical adjustment, measurement error or misclassification, recall bias, and different populations or study designs. 

The prevalence of individual-level risk factors that can change (i.e., increase or decrease) over time can be evaluated for their impact individually and collectively on shifting breast cancer stages at diagnosis in a population setting. The comprehensive modelling of individual-level rather than neighbourhood-level factors associated with breast cancer stage at diagnosis requires only limited resources and time, unlike intervention studies. The primary aim of this proof of principle simulation study is to explore the shifting of breast cancer stage at diagnosis downward by modifying individual factors singly or collectively using simulated data. In addition, we also aim to evaluate factors that could be targeted in policy or prevention programs that collectively maximize the shift to stage I at diagnosis. 

## 2. Methods

This simulation study required input values for individual-level factors that are associated with breast cancer stage at diagnosis, obtained from an appropriate statistical model fit to data on women diagnosed with breast cancer. These input values were acquired from a study of women who were diagnosed with breast cancer after enrolling in ATP between 2000 and 2008 [14]. ATP is a prospective health cohort that began in 2000 in which participants aged 35–69 years of age completed baseline questionnaires on personal and family history of cancer, dietary intake, cancer screening participation, reproductive history, and demographic information, such as household income, education, and employment status. 

A total of 492 women who joined ATP and were subsequently diagnosed with an invasive breast cancer up to January 2018 were included (Figure 1). Most women were diagnosed with Stage I disease (51%), followed by 37% of the women with Stage II disease, then 10% with Stage III disease and 2% with Stage IV disease. The American Joint Committee on Cancer (AJCC) version 6 (2004–2017), 7 (2010–2017) and 8 (2018+) staging was used in this study [17,18,19]. The outcome of interest is stage at breast cancer diagnosis, which is ordered from lowest to highest (I–IV), resulting in ordinal-scale data. Estimates of the association between individual-level factors measured at enrolment and stage at breast cancer diagnosis were obtained in a previous study [20]. In brief, multivariable partial proportional odds (PPOs) models were utilized as well as random forests to identify individual factors associated with breast cancer stage at diagnosis. We categorized the factors from this PPO regression model from Wang et al. [20] as follows: (1) non-modifiable or not easily modifiable factors (age at diagnosis, household income, Elixhauser comorbidity index and breast cancer family history); and (2) modifiable or potentially modifiable factors. The estimated odds ratios from the multivariable PPO model for these individual-level modifiable factors that were statistically significant (*p* < 0.08) were selected for this study (Table 1, Appendix A). Additional details on the modelling approach and results for the analysis of real data can be found in Wang et al. [20].

### 2.1. Proportional and Partial Proportional Odds Models 

Two common regression models for ordinal response data are the proportional odds (POs) and partial proportional odds (PPOs) models. The proportional odds (POs) model assumes that each factor has the same effect across all levels of the ordered stage outcome variable. For each ordered level of stage, denoted by level *j*, the PO model is given as follows:(1)$${Stage}_{j}=\alpha_{j}+X_{i}\beta +\varepsilon$$
where β is a vector of unknown regression coefficients for the individual risk factors, $X_{i}$ is the matrix of individual risk factors for *n* subjects, $\alpha_{j}$ is the intercept for the *j*^th^ level of cancer stage and $\varepsilon \;\sim N(0,\;\sigma^{2})$ is the random effect. Now, the probability of the cancer stage being higher than stage *j* is given as follows:(2)$$P\left(Stage>j\right)=\frac{\exp\;(\alpha_{j}+X_{i}\beta +\varepsilon )}{1+\exp\;(\alpha_{j}+X_{i}\beta +\varepsilon )}$$

Equation (2) is the PO model which assumes that the effect of factor $X_{i}$ does not vary across the levels of stages. This means the odds for a factor modelled as stage I (*j* = 1) vs. stages II + III + IV is the same as those for stage I + stage II vs. stage III + IV, etc. However, this assumption is not always realistic. For example, several studies have found that the probability of diagnosing cancer at a higher stage is greater for older individuals than younger ones, contradicting the POs assumption [21,22]. Wang et al. [20]’s study found that four factors (age at diagnosis, having been pregnant, spending time in the summer sun and currently experiencing stressful situation) did not satisfy the POs assumption. Therefore, they used the partial proportional odds (PPOs) model which has the following form:(3)$$P\left(Stage>j\right)=\frac{\exp\;(\alpha_{j}+X_{i}\beta +Z_{i}\gamma_{j}+\varepsilon )}{1+\exp\;(\alpha_{j}+X_{i}\beta +Z_{i}\gamma_{j}+\varepsilon )}.$$

In Equation (3), ${X_{i}}^{T}\beta$ is the contribution of the factors with equal slopes (β) across all four levels of the stage outcome, and $Z_{i}^{T}\gamma_{j}$ is the contribution of the factors with different slopes ($\gamma_{j}$) for each level *j* of the stage outcome. Both POs and PPOs regression models were used in our simulation study.

### 2.2. Data Simulation

Seven lifestyle factors that are potentially modifiable in a population and associated with breast cancer stage at diagnosis were considered in this study (see Appendix A for factor definitions). Protective factors from late stage at diagnosis included parity, spending time in the sun, having had a mammogram and high daily protein intake; however, increased risk of later stage at diagnosis included factors such as stress levels, high caloric intake and the use of birth control pills [20]. Table 1 describes these selected lifestyle factors, their individual effects on breast cancer stage and their distributional properties of mean and standard deviation (not median and quartiles as in Wang et al. [20]) in the study sample. 

The distributions of these selected lifestyle factors from the dataset for the 492 ATP women diagnosed with breast cancer provided the inputs needed for generating the simulated data. For the continuous dietary factors (total dietary energy intake and total dietary protein intake), 1000 samples were simulated using a truncated multivariate normal distribution that used the estimated means, correlations and standard deviation parameters from the real data set. A thousand samples for each binary variable were simulated using a Bernoulli distribution based on the estimated probabilities and pairwise correlations from the real data set. Distributions of the simulated data were checked by comparing summary statistics with the summary statistics of observed data from the ATP study and found to be similar (Appendix A). The values of the regression coefficients βs and $\gamma_{j}s$ as well as intercepts $\alpha_{j}$ for each *j* were obtained from the estimates found in Wang et. al. [20]. Since the proportion of women diagnosed at stage IV was very small, stages III and IV were combined in our simulation study.

### 2.3. Stage Shifting

Three different scenarios were evaluated in the simulation study. In Scenario I, each of the seven selected lifestyle factors identified by Wang et al. [20] were individually varied over five possible values, including the mean or proportion obtained from their study of real data. The remaining variables not being shifted were fixed at the observed value from the aforementioned study. In Scenario II, the factors that shifted stage at diagnosis to stage I by at least 0.5% in Scenario I were modelled using cumulative not individual effects. Lastly, Scenario III evaluated four factors (total dietary energy intake, total dietary protein intake, having had a mammogram and spending time in the sun from 11 am to 4 pm, June–August) that could be targeted in policies regardless of their impact on the shift downwards, also using cumulative effects.

Changes in the mean levels of the dietary factors from the values observed in the ATP study data were based on the effect of the factor. Higher-fat diets have been associated with weight gain, and previous studies show that excess weight is a significant risk factor in breast cancer aetiology [23]. Therefore, total dietary energy intake levels were modelled using the mean value of 1605 kcal/day, as well shifting it higher and lower by 100 and 200 kcal/day in the first scenario (Design 1, Table 2). Taking the value of 64 g/day of protein as the baseline, four increasing values of average daily total protein consumption by 5 g/day increments were evaluated in Design 2. Only increasing values of average daily total protein consumption were considered as higher levels as recent evidence suggests higher recommended daily allowances can improve one’s heath [24]. 

Ever having a mammogram was found to be protective as mammograms can detect breast cancer in early stages [25]. In Ontario, a mammogram cancer screening report reported a decline from 2012 to 2018 in cancer screening participation rates (83% to 66%) [26]. Although participation rates are based on the percentage of eligible women having had a mammogram in a 30-month period and not having ever had one in their lifetime, as measured in this study, our surrogate variable can provide insights into possible participation trends. Therefore, it is reasonable to check the effect of both increasing as well as decreasing the percentage of women having ever had a mammogram by 5–10%; this was evaluated in Design 3. Additionally, women who are pregnant at a younger age and have more children have a reduced long-term risk of breast cancer because of hormone-induced genetic changes in mammary glands, allowing mature breast cells to protect against breast cancer [27,28]. Women who are pregnant at an older age or women with low parity have a higher risk of breast cancer from increased oestrogen exposure, a hormone which can promote breast cancer [21,22]. Information about the ages of women when they were pregnant was not available in the ATP data. Therefore, increased proportions of 0.05 and decreased proportions of 0.05, 0.10 and 0.15 from the observed value of 0.90 were evaluated in Design 4 as women now tend to have children later in life and fewer children (e.g., declining birth rate) [29,30].

Several previous studies found that regular sun exposure is associated with substantial decreases in death rates from certain cancers. A systematic review found that while increased sun exposure is detrimental for cutaneous cancers, it may be protective for certain cancers through the production of vitamin D from sun exposure [31]. For breast cancer, there is a potential protective effect on aetiology [32]. Thus, it is reasonable to consider changing the amount of sunlight exposure, so we considered five distributions taking the observed distribution for the middle of the design matrix and changing it by 10% (see Design 5 in Table 2).

Chronic stress can negatively impact health by weakening the immune system which can promote tumorigenesis [33]. Chronic stress is a modifiable factor, which can be mitigated through various stress management methods [34]. Therefore, we considered the effects of decreasing or increasing the proportion of people experiencing at least one stressful situation in Design 6. Lastly, the proportion of women using birth control pills for any reason was also varied above and below the observed proportion of 0.87 who had taken them. A recent study reported that the use of oral contraceptives amongst Canadian women dropped by almost 20% between 2006 and 2016 [35]. Design 7 varied the proportion of women who had ever taken the pill. 

Scenario II evaluated the individual factors from Scenario I that shifted stage lower by at least 0.50% (Table 3. Designs 1–4, 7), as this could decrease the prevalence of late-stage disease by about 20%. Here, we only considered three values that shifted the stage at diagnosis proportions downwards towards the lower stages (see Table 4). Lifestyle factors were introduced one at time, building sequentially by adding each new factor to the previous ones in model. The final design, Design E in Table 4, represents the cumulative effects of all five factors fixed at the shift amounts denoted in Table 3. The final scenario, III, evaluated these four factors regardless of their impact to shift stage at diagnosis: total dietary energy intake, total dietary protein intake, having ever had a mammogram and spending time in the sun from 11 am to 4 pm, June–August. These factors could be potential policy or guideline targets so they were evaluated in this final scenario.

We simulated 20 realizations of the covariates and response for each combination of the distributions and generated 1000 samples per realization. The proportions of stages I, II and III were calculated, and median and the interval between lower and upper 2.25% of the ranked proportions (95% percentile interval (PI)) of the stages from 20 realizations were obtained. The R code used to generate the lifestyle factors and predict stage at diagnosis in our simulation study can be found in Appendix A. 

## 3. Results

The percentage of ATP women diagnosed at stage II and higher (49%) is very similar to Alberta population estimates (52%) for women [36]. These women have access to breast cancer screening without a referral from a doctor if they are aged 50 or older, so the barriers experienced by women in other jurisdictions are minimized. About 20% of the ATP women were less than 50 years of age when they were diagnosed with breast cancer, although most of them reported having had a mammogram previously (60.9%). The percentage of women between 40 and 49 years of age with a positive family history of breast cancer was 18.4%. However, other reasons for having had a mammogram previously are unknown for all participants. 

Table 3 shows the average percentage change, either up (+) or down (−), for each cancer stage when one lifestyle factor is changed from its observed value in the ATP data set. The other six factors are fixed at their observed values from the ATP data set in this scenario. For example, in Design 1, if the total dietary energy intake is decreased by 100 kcal/day, it results in 0.83% more women being diagnosed at stage I and fewer women diagnosed at stage II (−0.33%) and stages III and IV (−0.41%). When the total dietary energy intake and other factors are fixed to their observed ATP data set values and the total dietary protein is increased by 5 g/day to 70 g/day, this results in 2.18% more women being diagnosed at stage I (Design 2). Although modifying the proportion of people currently experiencing stressful situations or spending more time in the sun from 11 am to 4 pm June–August did not increase the proportion of women diagnosed at stage I, both did substantially reduce the proportion of women diagnosed at stages III and IV. Generally, increasing protective factors and decreasing risk factors shifted more women being diagnosed with early-stage disease downwards, with the greatest effect being seen by modifying dietary protein intake. Figure 2 illustrates these single-factor effect changes graphically.

The factors that singly increased stage at diagnosis by at least 0.5% were then evaluated by modifying their values (means or proportions) to increase their protective effect or decrease their risk effect (Table 4 and Figure 3). Beginning with total dietary energy intake, three lower shifted values were evaluated with the remaining four factors fixed at their observed ATP values (Design A). It was then fixed at 1400 kcal/day in the subsequent designs because that value resulted in the greatest increase at stage I diagnoses. Next, the total dietary protein was varied with the remaining three factors fixed at their observed ATP values in Design B. In Design C, the proportion of women having ever had a mammogram was varied from 0.83 to 0.95 with the total dietary protein now fixed at 85 g/day (optimal value) and the remaining two factors fixed at their observed values. With the total dietary intake fixed at 1400 kcal/day, the total dietary protein fixed at 85 g/day and the proportion of women having ever had a mammogram fixed at 0.95, the proportion of women who had ever been pregnant was increased while the proportion of women using birth control was fixed at 0.87 in Design D. Lastly, the first four factors were fixed at their optimal values and the proportion of women who had ever used birth control pills was decreased (Design E). Figure 3 illustrates the cumulative effects of increasing the proportion of breast cancer cases at stage I by optimizing these five factors and the corresponding decrease in the proportion of women diagnosed at breast cancer stages II, III and IV. Total dietary protein intake had the greatest single effect while the cumulative effect of additional changes in risk factors did not substantially increase the proportion of women diagnosed with early disease.

The results for Scenario III, concerning shifting lifestyle factors that could be potential policy targets, are given in the additional materials section (Appendix A). Using the same modelling strategy as that in the second scenario, the cumulative effects from adding each factor to the model at its most impactful level while fixing the others also showed that increasing the total dietary protein intake to 85 g/day had the biggest impact. Appendix A shows these results graphically. 

## 4. Discussion

This simulation study focused on modifying selected lifestyle factors associated with the stage of breast cancer at diagnosis to shift the cancer stage at diagnosis downward. Data were simulated based on lifestyle factors and their corresponding odds ratio estimates were obtained by Wang et. al. [20] using partial proportional models. Factors that increased the risk of early-stage diagnosis or those that decreased the risk of late-stage diagnosis were evaluated, the former by increasing its prevalence in the population and the latter by decreasing them. Both individual and cumulative effects were considered, including those that could be targeted in cancer prevention programs or policies. Increasing the total protein intake by 5 g/day on average in a population had the greatest effect on both increasing the proportion of breast cancer diagnoses at stage I and on decreasing the proportion diagnosed at stage III and IV. The next most impactful factor was total dietary energy intake for which a reduction of 100 kcal/day also led to a shift in the cancer stage downwards. 

Dietary factors have been previously identified with the risk of developing breast cancer or with impacts on survival [37]. A recent meta-analysis suggests that higher-fat diets are associated with an increased risk of breast cancer [23]. There is strong evidence that oestrogen level is associated with breast cancer risk and that increased dietary fat may increase the production of endogenous oestrogen [15,23]. Our results found that increasing total dietary energy (caloric) intake can increase the stage at breast cancer diagnosis, suggesting that low-fat diets could be beneficial in reducing one’s breast cancer risk and stage of cancer diagnoses. Results from the Nurses’ Health Study cohort of 6348 women diagnosed with stage I to III breast cancer concluded that there was a modest survival advantage with higher intakes of protein [38]. However, overall, biomarker-calibrated total protein intake was not associated with breast cancer incidence or mortality in another recent study [39]. Some studies suggest that higher vegetable protein intake was associated with lower breast cancer risk while higher animal protein intake was associated with an increased risk [39]. Our results also suggest that increased dietary protein intake is associated with early-stage breast cancer diagnoses; however, the model is limited to total intake and cannot evaluate vegetable protein and animal protein intake individually. There are potential mechanisms linking body fat and oestrogen levels with breast cancer risk but how these factors are associated with diet and physical activity requires substantially more research [15,23]. A recent systematic review and meta-analysis suggests that spending greater than one hour a day in the sun during the summer months could reduce the risk of breast cancer development [32]. However, few studies in the review accounted for skin type and sunscreen use [32]; studies which did account for these effects did not determine a significant effect [32]. Our results did not find a significant effect of outdoor sunlight exposure on lowering the stages of breast cancer at diagnosis, but skin type and sunscreen use were not incorporated as these were not available. 

Breast density and BMI are also factors which may affect breast cancer stage at diagnosis. A recent review found that denser breasts can mask cancers in mammograms due to only a 50% sensitivity for the highest breast density class [40]. As a result, there is a lower mortality reduction in women with dense breast tissues, and they have potentially increased chances of breast cancer detection at later stages [40]. However, this is not a modifiable risk factor nor was it available in the cohort data; thus, it was not included. In a recent Korean study, increased BMI was associated with advanced stage breast cancer diagnosis: 41% of people with obesity had breast cancer stages T2 to T4, compared to 28% and 23% in patients with a BMI of 25–29 or <25, respectively [41]. The Korean study also determined that obesity was potentially associated with a reduced likelihood of having palpable tumours [41]. While BMI was not statistically significant in the analysis of real data and so was not evaluated in this study, total dietary caloric intake may represent an individual’s potential BMI, based on average caloric requirements by age but not physical activity levels. Therefore, future studies should consider breast density, BMI and other stage-related factors.

While other early-stage breast cancer models exist, they differ in the factors included in their models [42,43]. A recent early breast cancer staging model utilized mammography and biopsy data to predict the stage at diagnosis; however, while that model used data from cancer screening or diagnostic procedures, it did not suggest how to reduce the risk of breast cancer being diagnosed at late stages [42]. Similarly, Cancer Intervention and Surveillance Modeling Network (CISNET) models also do not utilize lifestyle factors in addition to screening and screening-related factors to determine breast cancer risk [43]. Furthermore, the models determine risk reductions in incidence and mortality but do not focus on breast cancer stage [43]. The approach taken in our proof of principle study incorporated modifiable lifestyle factors, which enabled the effects of specific factors associated with breast cancer stage to be systematically evaluated. 

An important advantage of our modelling approach is its flexibility—it can be adapted for various cancer screening and prevention programs that focus on targeted screening and prevention messages. Another strength of this study is that the combined effects of two or more factors were modelled, as in real-life scenarios in which individuals may be affected by multiple factors. The single or cumulative effects of multiple factors could also be evaluated. Furthermore, the outcome, stage at diagnosis, was modelled on an ordinal scale rather than combining stage categories into early versus late, thereby potentially providing greater insights into factors that vary in their association across stage levels. A final strength of this simulation study is that our results could be obtained quickly and at a very low cost, compared to large population interventions that typically focus on a single factor. 

However, this study also has some limitations. The estimated ORs for the significant lifestyle factors were based on a single study utilizing data from one province, so they may not be similar to those of other populations, reducing the simulation’s generalizability. Although this simulation study approach could utilize various factors for various cancers, the focus was only on breast cancer in women. Some factors cannot be modified after the fact, such as having ever been pregnant, although changes in a population over time can occur. In addition, total dietary protein intake was only evaluated with increasing consumption as that would lead to better health, so the effects of decreasing consumption were not considered. The simulation model met most of the components of the STRESS guidelines, although additional validation and verification would be needed prior to implementing new cancer prevention programs [44]. Additional validation approaches and a decision-making framework could be used to verify and validate this modelling approach with specific validation metrics considered beyond the ones utilized here [45]. Lastly, some factors could be measured differently in other studies or with more granularity, such as the source of dietary protein. 

## 5. Conclusions

Reducing the diagnosis of late-stage breast cancer, that is, shifting the stage at diagnosis downwards, will naturally lead to reductions in mortality rates. This is an important benefit to diagnosing breast cancer—or most cancers—at an early stage. Our proof of principle modelling approach enables cancer prevention programs to evaluate the potential impact of changes in modifiable lifestyle factors in a population and then develop interventions that target those factors. If lifestyle factors that increase the risk of late-stage diagnoses are becoming more prevalent in a population, then cancer control programs can prepare for the potential increase in late-stage diagnoses and devise strategies to mitigate them. Policies could also be created or modified too. Other researchers can generalize this approach to factors beyond lifestyle and to different types of cancer. Identifying the most influential factors that contribute to breast cancer stage at diagnosis, especially those that are modifiable, continues to be relevant for cancer risk reduction. 

## Figures and Tables

**Figure 1 cancers-16-01201-f001:**
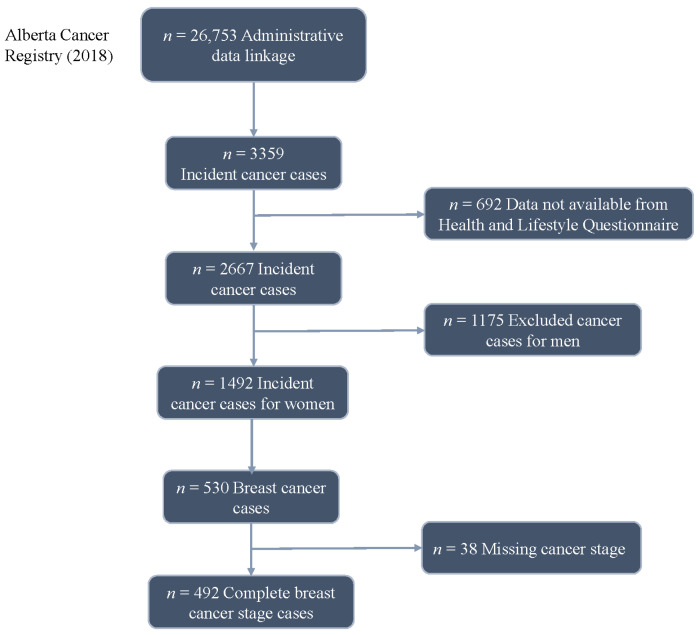
Study population selection from ATP participants who were diagnosed with breast cancer (reproduced from [20]).

**Figure 2 cancers-16-01201-f002:**
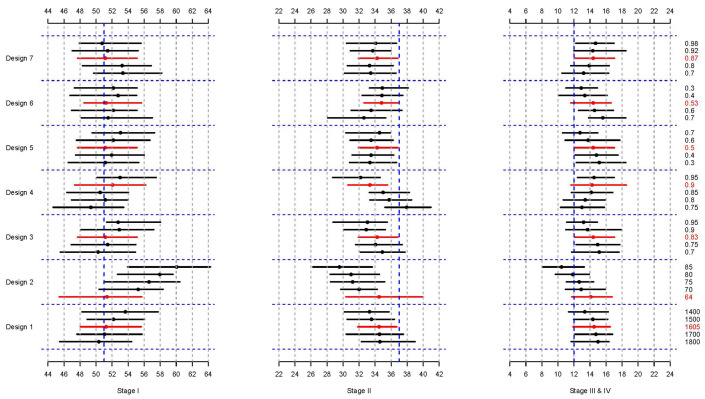
Impact of modifying the distributions for each lifestyle factor while fixing the other six factors at their observed values in Scenario I. The medians (dots) and 95% percentile intervals (PIs, horizontal lines) of the observed (red) and predicted (black) breast cancer stages after the modifications are shown. The blocks between the blue horizontal dotted lines denote the effects of each factor. The vertical black dotted lines represent 2% while the vertical blue dashed lines represent the percentage of women diagnosed in that stage in the real data set.

**Figure 3 cancers-16-01201-f003:**
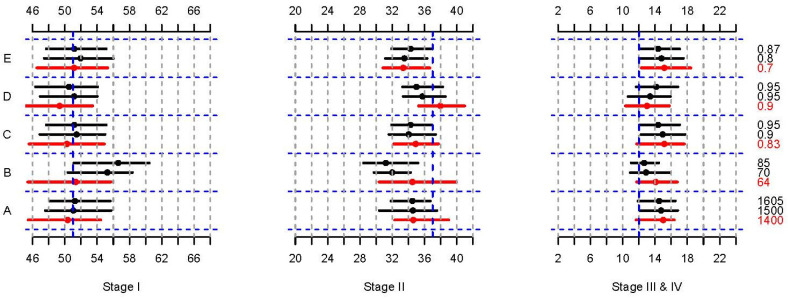
Cumulative effects of modifying the distributions of the factors from Designs A to E and setting them to their optimal values for most significant lifestyle factors in Scenario II. The medians (dots) and 95% percentile intervals (PIs, horizontal lines) of the observed (red) and predicted (black) breast cancer stages after the modifications. The blocks between the blue horizontal dotted lines denote the effects of each factor. The vertical black dotted lines represent 2% while the vertical blue dashed lines represent the percentage of women diagnosed in that stage in the real data set.

**Table 1 cancers-16-01201-t001:** Properties of the selected modifiable factors from the ATP data set (N = 492).

Lifestyle Factor	Effect *	Mean	Standard Deviation
Total dietary energy intake (kcal/day)	Increased risk	1605	576
Total dietary protein intake (g/day)	Decreased risk	64	25
		N	%
Have had mammogram test (Yes)	Decreased risk	409	83.1
Have been pregnant (Yes)	Decreased risk	444	90.2
Spend time in the sun between 11 am and 4 pm from June to August (≥1 h/day)	Decreased risk	248	50.4
Currently experiencing stressful situation (≥1)	Increased risk	263	53.5
Use birth control pills (Yes)	Increased risk	427	86.8

* Effect: Odds ratio for increased risk factor > 1 and for decreased risk (protective factor) < 1 for later stage at diagnosis.

**Table 2 cancers-16-01201-t002:** Modified means and proportions of the lifestyle factors for Scenario I.

Design	Factor	Modified Distribution	* ORs (95% CI)
1	Total dietary energy intake (numeric, kcal/day)	(1800, 1700, **1605**, 1500, 1400)	1.07 (1.01–1.14)
2	Total dietary protein intake (numeric, g/day)	(**64**, 70, 75, 80, 85)	0.98 (0.97–1.00)
3	Ever had a mammogram (proportion)	(0.7, 0.75, **0.83**, 0.90, 0.95)	0.57 (0.33–0.97)
4	Ever been pregnant (proportion)	(0.75, 0.80, 0.85, **0.9**, 0.95)	0.51 (0.27–0.98)
5	Spending time in the sun from 11 am to 4 pm, June–August (proportion)	(0.3, 0.4, **0.5**, 0.6, 0.7)	0.56 (0.32–0.99)
6	Currently experiencing stressful situations (proportion)	(0.7, 0.6, **0.53**, 0.4, 0.3)	2.87 (1.55–5.32)
7	Using birth control pills (proportion)	(0.70, 0.80, **0.87**, 0.92, 0.98)	1.64 (0.94–2.86)

* ORs (95% CI) are considered from Wang et. al. [20] Effects: increased risk if OR > 1 and decreased risk if OR < 1. Observed means and proportions from ATP study data in bold, underlined text.

**Table 3 cancers-16-01201-t003:** Average percentage change for each cancer stage after modifying the lifestyle factors’ distributions in Scenario I by the shift amount and the effect of that shift on late-stage risk.

Design	Factor	Shift Amount	Average % Change			Effect
			Stage I	Stage II	Stage III and IV	
1	Total dietary energy intake (numeric, kcal/day)	−100 kcal/day	0.83	−0.33	−0.41	Increased Risk
2	Total dietary protein intake (numeric, g/day)	+5 g/day	2.18	−1.24	−0.91	Decreased Risk
3	Having ever had a mammogram (proportion)	+5%	0.63	−0.46	−0.49	Decreased Risk
4	Having ever been pregnant (proportion)	+5%	0.91	−1.44	0.38	Decreased risk
5	Spending time in the sun between 11 am and 4 pm, June–August (proportion)	+10%	0.48	0.30	−0.60	Decreased Risk
6	Currently experiencing stressful situations (proportion)	−10%	0.16	0.58	−0.68	Increased Risk
7	Using birth control pills (proportion)	+10%	−0.65	0.15	0.38	Increased Risk

**Table 4 cancers-16-01201-t004:** Modified distributions of the five most significant lifestyle health factors from Scenario II, their shifted values and the effect of these shifts on late-stage risk.

Design	Factor	Shifted Values	Effect
A	Total dietary energy intake (kcal/day)	(1400, 1500, **1605**)	Increased Risk
B	Total dietary protein intake (g/day)	(**64.0**, 70.0, 85.0)	Decreased Risk
C	Having ever had a mammogram	(**0.83**, 0.90, 0.95)	Decreased Risk
D	Having ever been pregnant	(**0.90**, 0.95, 0.95)	Decreased Risk
E	Using birth control pills	(0.70, 0.80, **0.87**)	Increased Risk

Observed means and proportions from ATP study data in bold, underlined text.

## Data Availability

Access to individual-level data is available in accordance with the Health Information Act of Alberta and Alberta’s Tomorrow Project (ATP) Access Guidelines at https://myatpresearch.ca/DataAccess (accessed on 14 October 2022).

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
