# Peer review of "Stage Shifting by Modifying the Determinants of Breast Cancer Stage at Diagnosis: A Simulation Study"

_cancers, 2024, doi:10.3390/cancers16061201_

Round 1

Reviewer 1 Report (Previous Reviewer 4)

Comments and Suggestions for Authors

The manuscript may be accepted for publication.

Reviewer 2 Report (Previous Reviewer 2)

Comments and Suggestions for Authors

The article has been modified and added practical significance. The introduction and conclusion have been rewritten taking into account the comments of the reviewer.

Comments on the Quality of English Language

The article has been modified and added practical significance. The introduction and conclusion have been rewritten taking into account the comments of the reviewer.

Reviewer 3 Report (Previous Reviewer 5)

Comments and Suggestions for Authors

The paper "Stage shifting by modifying the determinants of breast cancer stage at diagnosis: a simulation study" is considered a subject of great interest to the scientific world and a subject of current interest. The manuscript is very interesting, well-written, and easy to understand.

This manuscript is a resubmission of an earlier submission. The following is a list of the peer review reports and author responses from that submission.

Round 1

Reviewer 1 Report

Comments and Suggestions for Authors

Introduction

Page 2, lines 78, 79 – does access to free breast cancer screening without the need for referral reduce the impact of socioeconomic factors – what is the evidence for this?  I believe there is plenty of evidence to suggest that attendance at screening is related to socioeconomic factors in areas that offer free breast cancer screening e.g. the UK.

Methods

492 with breast cancer – how many women were recruited to the study? Figure 1 is missing.

When was risk factor information in the ATP study collected – at baseline only?

Page 3, line 147, Table 1 – can ever having been pregnant really be considered as a modifiable risk factor?

Page 4, line 169 – it would be useful to have a table showing the real and simulated data.

Page 5, line 185 – which four factors?

Page 5, line 184 – is there a justification for the choice of 0.5% for stage shift?

Page 5, line 194 – is there a reason why decreasing protein g per day was also not considered?

The methods are hard to follow and could be written more clearly.

Results

Similarly the results are hard to follow and could be written more clearly.

Many women in the study were under screening age (50 years) is this likely due to a family history of the breast cancer – can you say anything about this?

It would be helpful to report 95% CIs or percentiles for changes in downward shifts in the Tables – are the changes statistically significant?

Figure 2 & 3 – add a footnote to describe what is presented in the Figure i.e. what is in written in page 8, lines 278-291.  Do the black lines represent each modification – can you label each line within the design to show these values? Say what the vertical blue dotted line is at each stage – presumably it’s the percent in each stage from actual data?

Discussion

There are likely to be other risk factors that are associated with stage of breast cancer e.g. breast density – although not necessarily easily modifiable, and further research is required, chemoprevention drugs may be beneficial for reducing mammographic breast density. This should be discussed. BMI is also likely to be important and linked to some of the factors in this study.

Can you say anything about bias in those registering on the ATP study – are they more likely to be less socioeconomically deprived?

Comments on the Quality of English Language

Minor

Page 1, line 22 – the following part of the sentence requires rephrasing ‘Previous simulation study models to shift breast cancer stage at diagnosis lower…’

Page 1, line 24 – ‘maybe’ should be ‘may be’

Page 1, line 26 – ‘data-based’ should be ‘database’

Page 8, line 299 – ‘bseast’ should be ‘breast’

Reviewer 2 Report

Comments and Suggestions for Authors

The article refers to current epidemiological studies that can shed light on new approaches in the diagnosis and development of malignant neoplasms. It is noted that the study of risk factors, the prognosis of the disease can change the picture of the disease, affect economic indicators and life expectancy. From the standpoint of evidence-based medicine, the authors show the shifts in staging associated with breast cancer.

Comments on the Quality of English Language

The article refers to current epidemiological studies that can shed light on new approaches in the diagnosis and development of malignant neoplasms. It is noted that the study of risk factors, the prognosis of the disease can change the picture of the disease, affect economic indicators and life expectancy. From the standpoint of evidence-based medicine, the authors show the shifts in staging associated with breast cancer. However, the article practically does not show the significance of the work, what it gives to the community as a whole.

Reviewer 3 Report

Comments and Suggestions for Authors

Rewrite key words based on the Mesh.

This doi: 10.3389/fonc.2022.921015. eCollection 2022. Can be useful for introduction

In introduction, gap and purpose should be expressed in better sentence. Also All sentences need references. Line 57, 68 and…

What about internal and external validation of the model?

How is it tested on real data?

In the discussion, all the sentences need a reference.

What are the items included in the final model?

In the discussion, all the sentences need a reference. For example line: 369 -384??

The conclusion is long, and needs no references.

State the limitations.

Reviewer 4 Report

Comments and Suggestions for Authors

1.  The authors have not provided any validation after their identification of determinants of stage shifting through simulation study.

2. Why was the proportional odds (PO) model chosen by the authors? Bayesian approach shows better results.

3. No Figure 1 is provided in spite of its mention in the manuscript.

4. The authors should discuss in what way their observation differs from Wang et al (reference: 15) as the parameters chosen were same. 

5. Authors are requested to mention all the risk factor they consider for this study.

6. In line 162-164, authors stated that they had 492 actual samples and they simulated 1000 samples, however, the data presented here is based on the final 1492 samples? Do the authors need to increase the sample simulation numbers.

Reviewer 5 Report

Comments and Suggestions for Authors

The paper "Stage shifting by modifying the determinants of breast cancer stage at diagnosis: a simulation study" is considered a subject of great interest to the scientific world and a subject of current interest. The manuscript is very interesting, well-written, and easy to understand.

I advise the authors to review the following points in the manuscript, as in the pdf file, there seem to be errors:

Line 55, 144, 201, 206, 234, 298, 309, 321, 346, 353, 366, 367, 370, 371, 373, 377, 387: The pdf file seems to have a double spacing. Check the spaces and eventually correct the text.

Line 108/109: In the pdf file, there seems to be a different character in the text, bigger than the other text.

Line 118: Do not see figure 1.

Line 125: in the text missing a symbol (where ??? is a vector)

Line 173: (15) Since = (15). Since

Line 195: 5g/day = 5 g/day

 Line 229 - Table 2: Arranging spaces and commas between numbers.

Line 243: Scenario I.. = Scenario I.

Table 3: Missing effect in drawing four (4). What is the effect?

Line 251: in in = in

Figure 2 / 3 = What is the unit of measurement of the numbers in the drawing? 

Line 299: bseast = breast